# The Impact of Location-Based Tax Incentives and Carbon Emission Intensity: Evidence from China’s Western Development Strategy

**DOI:** 10.3390/ijerph20032669

**Published:** 2023-02-02

**Authors:** Yufeng Wang, Shijun Zhang, Luyao Zhang

**Affiliations:** 1Business School, Foshan University, Foshan 528000, China; 2Research Centre for Innovation & Economic Transformation Research, Institute of Social Sciences in Guangdong Province, Foshan 528000, China; 3School of Finance and Trade, Wenzhou Business College, Wenzhou 325035, China; 4School of Business, Hanyang University, 222 Wangsimni-ro, Seongdong-gu, Seoul 04763, Republic of Korea

**Keywords:** carbon emission intensity, regional development, place-based policies, Western Development Strategy (WDS), difference-in-differences method

## Abstract

This study seeks to address the question of whether China’s Western Development Strategy (WDS) has affected the carbon emission intensity of the regions it covers. There remains a distinct lack of analysis based on the normative causal inference method regarding the impact of this economic development policy on carbon emissions. Our research contributes to the large body of international literature studying the effects of place-based policy and has implications for place-based policies regarding the impact of carbon emissions. It constructs a duopoly model to illustrate the relationship between lower prices of capital (caused by policies such as tax reduction) and carbon emissions. Using county-level data on both sides of the provincial boundary of the WDS from 1998 to 2007, and applying the difference-in-differences method, our results indicate that the WDS has significantly increased carbon emission intensity of the western counties. Our findings also indicate that while the WDS has had no significant positive effect on counties’ economic growth, no policy trap effect was found. There is also no evidence suggesting that the economic activities attributable to the WDS have brought any negative externalities of carbon emissions to the counties east of the western provincial border.

## 1. Introduction

Carbon emissions and climate change are major issues facing humanity in the 21st century. They are also strategic issues that need to be actively addressed in the process of China’s economic and social development. As one of the members and advocates of the United Nations Framework Convention on Climate Change (1992) and the Kyoto Protocol (1997), China has specifically incorporated the goal of reducing carbon intensity into its national economic plan [1]. However, China has not yet fully realized urbanization and industrialization, and the level of social welfare nationally remains somewhat low. China must aim to achieve economic growth while simultaneously reducing carbon dioxide emissions. To accomplish this goal while achieving sustainable economic and social development, it is necessary to have a more in-depth and detailed understanding of the various behaviors and underlying mechanisms related to carbon emissions.

Regional economic development policy is an important means of promoting national economic development [2,3]. Since China commenced its policy of reform and opening up in 1978, regional economic policies have played an important role in the rapid development of China’s economy [4]. To achieve long-term sustainable development, the implementation of region-specific economic policies will remain essential. The final effect of the policy will reflect all aspects of social and economic operations inside and outside the region [5]. The WDS is arguably the single most important Chinese regional economic policy in this century. Researchers have evaluated this policy from different dimensions, and there are still debates on its effectiveness. Some studies have affirmed the positive impact of the WDS on economies, asserting that it has alleviated the inequality of regional development in China [6] and was vital to the country’s handing of the energy crisis [7,8]. However, some researchers hold the opposite view that the WDS did not play its intended role as a growth driver, but as a “policy trap” [9,10]. For instance, the fast pace of targeted development requires a vast amount of energy, leading to enormous CO_2_ emissions [11,12]. Does the WDS promote or hinder the economic development of the western region? Is the impact of the WDS on the surrounding areas positive or negative? There is still no definite answer to these questions.

Some research shows that the regional economic development strategy has promoted local urbanization and related income and lifestyle changes [13,14]. There are also studies illustrating that urbanization and industrialization greatly contribute to the growth of carbon emissions [15,16]. Has the WDS directly led to an increase in local carbon emission intensity? Does the impact of the increased intensity of carbon emissions offset the positive role of the WDS in promoting local economic development? To understand these issues, it is necessary to comprehend the internal logical relationship between the WDS and the intensity of carbon emissions. This study uses county-level data of the eastern and western neighboring provinces on both sides of the WDS boundary, and applies a technique similar to the double difference method to examine whether the WDS has increased the carbon emission intensity and total carbon emissions of the counties on the west side of the provincial boundary. Based on the relevant test results of this study, the issues related to the evaluation of the WDS are then extended and discussed [17,18]. A key contribution of this study is that by using econometric methods that are more reliable, it confirms the WDS has improved the carbon emission (intensity) of China’s Western region. The specific objective of this study is to understand the relationship between the WDS and carbon emissions, by measuring the role of tax reduction policies in the production and capital expansion decisions of enterprises within the WDS area.

The primary research goal is to find out how the WDS affects the behavior of economic entities in this region, and how this in turn affects total carbon emissions and related economic performance. This study simultaneously aims to create a deeper overall understanding of the consequences of the WDS.

The remainder of this paper is organized as follows: Section 2 provides a review of the literature on the WDS and carbon emissions issues. Section 3 describes the theoretical models of the hypotheses to be tested. Section 4 outlines the methodology and results. Section 5 and Section 6 conclude by discussing the practical implications, limitations of this study, and potential future research directions.

## 2. Literature Review

A large amount of literature has been produced on the two major research fields of economic development policy and carbon emissions. Lei et al. [19] developed a carbon-weighted economic development indicator covering the dimensions of energy, environment, economy, and resources, based on the data envelopment analysis framework to determine the driving force of China’s carbon-emission-conscious economy. The research results show that changes in carbon emission intensity are brought about by the adjustment of foreign direct investment, local fiscal expenditure, industrial structure, and energy consumption structure. The carbon emission intensity of China’s GDP has decreased significantly this century, mainly due to the continuous decline of energy intensity in the production sector, and the concurrent fall in demand for direct energy consumption [20]. Wang et al. [21] measured Chinese provincial carbon emissions, and used the factor decomposition method to investigate the reasons for the changes. Their research shows that per capita GDP is decisive in inhibiting the growth of carbon emissions. The main factor is the decline in energy intensity, with this change deriving mainly from the reduced energy intensity in the industrial sector.

The relationship between the WDS, energy consumption, and carbon emissions has also received much attention recently. Liu et al. [22] used provincial panel data to examine the total factor energy efficiency of the two development policies, the WDS, and the Northeast Revitalization Plan. Their research showed that while the implementation of the WDS did not improve the energy efficiency of the western region, the execution of the Northeast Revitalization Plan has improved the energy efficiency of the Northeast region. Lu and Deng [23] found that the WDS brought about a decrease in the total carbon emissions of western provinces. However, the study used basic descriptive statistical analysis methods, and did not control for the annual effect or initial characteristics, which may lead to large estimation errors, making the conclusion worthy of further research and verification. Zhang et al. [24] used the propensity score matching difference-in-differences (PSM-DID) method using provincial panel data and found that the WDS has increased the carbon emission intensity of the western provinces. In general, due to the serious problem of endogeneity in the selection of treatment groups, the evaluation results at the provincial level may need to be supported by more grassroots (such as county-level) data research to be considered more robust.

Normative assessment of the WDS and carbon emissions can also provide a reference for bridging the controversy in the policy evaluation literature on the WDS. Two important, divergent themes remain in this area. One is whether the WDS has promoted the economic growth of the western region. Golley [17] estimated using provincial panel data that the WDS has increased the growth rate of western provinces. Warner’s [25] study using the data of counties on both sides of the provincial boundary of the WDS found that the WDS has significantly improved the GDP of western counties. However, this study used a provincial fixed effect model and did not adopt the DID method, thus casting doubt on the robustness of the results obtained. Dai et al. [26] used enterprise-level data and found that while the development of the western region improved enterprise total factor productivity (TFP); this effect had weakened over time. Liu et al. [27] applied the triple-difference method and found that the WDS has increased the number of enterprises in the counties on the west side of the western provincial boundary. The authors believe that this is because policy incentives have resulted in more entrepreneurial behaviors. Most of the literature in the field confirms the positive significance of the WDS at the economic level, at least in the short term. However, Shao and Qi [28] applied an econometric analysis of the relationship between the WDS and economic growth in China’s Western regions and the transmission mechanism between energy exploitation and economic growth using cross-provincial panel data collected from 1991–2006. Findings indicate that after the implementation of the WDS, energy exploitation impeded economic growth mainly through three indirect transmission channels: the crowding-out effect on human capital input, on science and technology (S & T) innovation, and the weakening of institutions through rent-seeking and corruption. In other words, the WDS has not promoted growth in, and has even become an “economic trap” for China’s western region.

Another topic of investigation is whether the WDS has brought “negative externalities” to neighboring eastern regions. Liu et al. [27] found that the number of enterprises in counties on the east side of the western provincial boundary was not significantly negatively affected. When studying the impact of local policies on economic development, Luo et al. [29] stated that enterprises in economically developed areas would migrate to less developed areas because of the “tax depression effect,” which constitute a negative external effect.

This brief literature review demonstrates that the relationship between the WDS, carbon emissions and other related elements is worthy of more in-depth and normative research. Therefore, this paper attempts to provide a simple theoretical explanation of the mechanism through which the WDS affects carbon emissions. County-level data is used to estimate the impact of the WDS on carbon emission intensity based on the causal inference idea, while the impact of the WDS on economic growth, and the corresponding “external effects” are also discussed. The marginal contribution of this paper may derive from the fact that county-level data estimates are more in line with the requirements of causal inference, thus making the estimation results more credible. The results of this study can thus provide a reference point for bridging the existing disputes over the economic growth effect and external effect of the WDS.

## 3. Materials and Methods

### Hypotheses and Research Model

To understand the relationship between the WDS and the quantity of carbon emissions, it is necessary to know which economic policies most affect economic enterprises in the WDS, how enterprises’ behavior changes as a result, and how this in turn affects the level of carbon emissions and related economic performance variables. The effects of the WDS are mainly reflected in the tax preferences for specific enterprises, with carbon emissions mainly generated from the energy consumption of enterprises [30]. To understand the relationship between the WDS and carbon emissions, it is necessary to clarify the impact of tax reduction policies on the production and capital expansion decisions of enterprises. In this section, we first briefly introduce the WDS, especially its preferential tax policy, and then build a brief model to explain the impact of tax reduction policy on the optimal output and capital of enterprises. On this basis, in combination with the actual situation of the WDS and relevant literature, this paper explains the possible impact of the WDS on carbon emissions and other economic performance variables (such as output), and puts forward the hypothesis and research model of this paper.

The purpose of the WDS is to reverse the unbalanced economic development of the eastern and western regions of China through a series of preferential policy supports; the most important of these is a substantial income tax preference given to enterprises in industries supported by the state [31]. (The main basis of the WDS tax policy is the “Notice of the General Office of the State Council on Implementing Several Policies and Measures for the WDS.” Among them, it is stipulated, “all policies and measures and their detailed rules shall be implemented from 1 January 2001.” Later, some documents made further clarification on the change of the catalog of tax preferential policies applicable to the WDS, but did not change the basic policies. According to the above policy, WDS has significantly reduced the income tax of eligible enterprises in the western region (from 25% to 15%, with a decrease of 54%), which is a very significant preference. The research of Luo et al. [29] studies tax changes in China and the result shows that WDS has reduced corporate income tax in the western region by 39.5%).

How will this tax preference affect the cost-benefit structure of the beneficiary enterprises and thus their production behavior? Corporate income tax incentives do not usually affect demand for enterprise products, and as wage expenses are deductible items, the tax reduction does not directly affect the price of labor [32]. In terms of the cost to enterprises, these income tax incentives somewhat lower the price of capital [33].

The carbon emissions of enterprises are mainly generated through fixed capital investment and production. Therefore, the WDS influences corporate decision-making, and thus, regional carbon emissions, in the same way that reduced prices for capital impact on the optimal capital stock and output of enterprises.

To understand this effect, we attempted to construct a duopoly output decision-making model. In this model, we assume that the technology and products of the enterprises are homogeneous, and the production function is assumed to be in the form of a Cobb-Douglas production function with constant returns to scale. Namely [34,35],
$$Q_{i}=k_{i}{}^{\alpha }L_{i}{}^{1-\alpha },\;i=1,2;\;\alpha \in \left(0,1\right)$$

At the same time, as it is assumed the demand curve faced by the enterprise is in the linear form: $Q^{d}=\beta -\lambda p$, we can get: $p=\left(\beta -\left(Q_{1}+Q_{2}\right)\right)/\lambda$. Assuming that the labor market of the two enterprises is perfectly competitive, the price is w. Due to the aforementioned subsidy, the capital prices are different, and are respectively $r_{1}$ and $r_{2}$. Thus, the cost functions of the two oligarchic enterprises are:$$TC_{1}=wL_{1}+r_{1}k_{1}\;TC_{2}=wL_{2}+r_{2}k_{2}$$

According to the cost minimization principle of firm 1, we can get:∂Q1∂k1∂Q1∂L1=r1w⟹L1=1−αα×r1wk1

So: $Q_{1}=k_{1}^{\alpha }L_{1}^{1-\alpha }=Gk_{1}$, where $G={ [\frac{\left(1-\alpha \right)r_{1}}{\alpha w}]}^{1-\alpha }$

Similarly, there are: $k_{2}=\frac{Q_{2}}{H},\;H={ [\frac{\left(1-\alpha \right)r_{2}}{\alpha w}]}^{1-\alpha }$,

The profit of firm 1 can be expressed as:$$\pi_{1}=PQ_{1}-wL_{1}-r_{1}k_{1}$$

That is: $\pi_{1}=\frac{\beta -Q_{1}-Q_{2}}{\lambda }Q_{1}-\frac{r_{1}}{\alpha G}Q_{1}$

From the first-order condition of firm 1’s profit maximization, we can get:$$\frac{\partial \pi_{1}}{\partial Q_{1}}=\frac{\beta -Q_{1}^{*}-Q_{2}}{\lambda }-\frac{Q_{1}^{*}}{\lambda }-\frac{r_{1}}{\alpha G}=0,\;\mathrm{that}\;\mathrm{is}\;Q_{1}^{*}=\frac{\lambda }{2}\left(\frac{\beta -Q_{2}}{\lambda }-\frac{r_{1}}{\alpha G}\right).$$

Similarly, for firm 2, there are: $Q_{2}^{*}=\frac{\lambda }{2}\left(\frac{\beta -Q_{1}}{\lambda }-\frac{r_{2}}{\alpha H}\right)$

Combining the above two formulas, we can get:$$Q_{1}{}^{*}=\frac{4}{3} [\frac{\beta }{4}-\frac{\lambda }{2\alpha^{\alpha }}\times {\left(\frac{w}{1-\alpha }\right)}^{1-\alpha }\times r_{1}{}^{\alpha }+\frac{\lambda }{4\alpha^{\alpha }}\times {\left(\frac{w}{1-\alpha }\right)}^{1-\alpha }\times r_{2}{}^{\alpha }]\;Q_{2}^{*}=\frac{4}{3} [\frac{\beta }{4}-\frac{\lambda }{2\alpha^{\alpha }}\times {\left(\frac{w}{1-\alpha }\right)}^{1-\alpha }\times r_{2}{}^{\alpha }+\frac{\lambda }{4\alpha^{\alpha }}\times {\left(\frac{w}{1-\alpha }\right)}^{1-\alpha }\times r_{1}{}^{\alpha }]$$

From the partial differential expression of firm 1’s optimal output decision expression to its capital price, we can get:

**Lemma 1** **(L1).***When the price of capital decreases, the oligopoly’s optimal output increases*. (The relevant parameters λ, β, w, $r_{1}$, $r_{2}$ are all greater than 0, $\frac{\partial Q_{1}^{*}}{\partial r_{1}}<0$.)

From this, we can further explore the relationship between capital price and optimal capital usage. The relationship between the two is obtained by the following formula:$$\frac{\partial k_{1}}{\partial r_{1}}=\frac{\partial Q_{1}{}^{*}/\partial r_{1}}{G\left(r_{1}\right)}-\frac{Q_{1}{}^{*}\left(r_{1}\right)}{G^{2}\left(r_{1}\right)}\;∵\frac{\partial Q_{1}{}^{*}}{\partial r_{1}}<0,\;G\left(r_{1}\right)>0,\;Q^{*}\left(r_{1}\right)>0\;∴\frac{\partial k_{1}}{\partial r_{1}}<0$$

That is,

**Lemma 2** **(L2).***The amount of capital used by an oligopoly is inversely related to its capital price, that is, as the capital price decreases, the optimal amount of capital increases*.

The relationship between the capital price of an oligarchic enterprise and the capital stock of another enterprise is expressed by [36,37]:$$\frac{\partial k_{1}}{\partial r_{2}}=\frac{\partial Q_{1}^{*}/\partial r_{2}}{G\left(r_{1}\right)}<0$$

That is,

**Lemma 3** **(L3).***If the capital price of an oligopolistic firm falls, the optimal amount of capital of other oligopolistic firms will fall. If the capital price of an oligopoly decreases, the optimal amount of capital of another oligopoly will decrease*.

Next, we will discuss the specific effects of the WDS in combination with the preliminary conclusions of the above model and the actual economic operation logic. According to Lemma 1 and Lemma 2, if the capital price of the beneficiary oligopoly decreases, its optimal capital quantity and output will both increase. (Limited by the preconditions of the model’s oligopolistic economic structure, the number of firms will not change. However, putting this assumption aside tax incentives attract new firms, which can also lead to an increase in output and capital stock). The main beneficiaries of the WDS are concentrated in mining, manufacturing, and other related industries. According to Yu [38], mining and manufacturing are the industries with the highest carbon emission intensity. This means average carbon emission intensity and total carbon emissions will increase if the output and capital stock of beneficiary enterprises are increased, assuming that the output of non-beneficiary enterprises remains unchanged. Even if the output of other industries decreases due to other reasons, as long as the total output remains unchanged, the average carbon emission intensity and total carbon emissions will still increase. Therefore, the core hypothesis of this paper can be drawn:

**Hypothesis 1** **(H1).***The WDS will increase the carbon emission intensity and total carbon emissions in the western region*.

This hypothesis can be tested by the following difference-in-differences estimating equation based on the panel fixed effects framework:$$Den_{it}=\alpha \times West_{t}\times Post_{t}+\gamma X_{it}+\mu_{i}+\nu_{t}+\varepsilon_{it}$$

Among them, the dependent variable is the county-level carbon emission intensity (or total). The dummy variable West represents whether it is a WDS county. The variable is assigned to 1 in the west and 0 in the east of the WDS provincial boundary. The dummy variable Post indicates whether it is the year after WDS implementation. The post-implementation year is assigned a value of 1 after 2000, and the other is 0. The product of these two variables is the core independent variable, and its estimated coefficient represents the carbon emission effect of the WDS. X is the vector of control variables. μ is the fixed effect of the county, and ν is the fixed effect of year. ε is the interference term, and the subscripts i and t represent county and year, respectively.

This research selects samples from counties on both sides of the eastern and western provincial boundaries. In order to evaluate the policy effect effectively and in a statistically unbiased manner, in principle, the policy treatment group and the control group should be indistinguishable or randomly selected. As the WDS is a targeted regional policy, the selection of policy objects is not random. To solve this problem, this paper uses the county, the area near the provincial boundary as the research unit. (The county-level administrative regions referred to in this article include counties, autonomous counties, county-level cities, and subordinate districts and banners of prefecture-level cities). The WDS is based on the division of provinces. Although the eastern and western provinces differ greatly in in terms of their geographical characteristics, other factors of the two counties on both sides of the provincial boundaries are often similar. Therefore, compared with using the whole sample, or using a larger geographical scope as the research object, using the counties on both sides of the provincial boundary in the east and west as the treatment and control group can more effectively meet the conditions of the difference-in-differences method.

According to Lemma 3, as enterprises in the west get preferential treatment, enterprises based in the east are relatively weaker in terms of competition, which may lead to a “redistribution” effect or “negative externality problem” on the eastern side, especially in neighboring regions. From the above equation estimates, the estimated policy effect may be (partly) due to the redistributive effect. To verify or exclude this effect, it is only necessary to replace the samples in the above equation with the counties to the east of the provincial boundary of the WDS, and to group them according to the different distances. In addition, according to Lemma 1, a decrease in capital price will lead to an increase in optimal output. If different types of industries have different carbon emission intensities, the increase in output will lead to an increase of carbon emissions in the same proportion. Considering that the preferential policies for WDS mainly benefit large-scale and industrial-oriented industries, the increase in carbon emissions brought about by the WDS may be greater than the proportion of the increase in total output. However, the tax reduction policy of the WDS may also be accompanied by further government intervention, leading to market segmentation, monopoly, and rent-seeking, reducing the effectiveness of market resource allocation [28], and the potential output level. Therefore, the total growth effect of the WDS depends on the “sum” of these two opposite effects, and the direction of the total effect is unclear. The research framework is shown in Figure 1. In response to the relevant debate, the following section will also examine the overall impact of the WDS on total output, and the issue of “external effects.”

## 4. Results

### 4.1. Data and Descriptive Statistics

The core explanatory variables of this paper are the county-level carbon emission intensity and carbon emission quantity. The carbon emission data used are from the county-level carbon emission data calculated by Chen et al. [39] based on nighttime light data. Considering the research purpose of this paper and the availability of relevant control variable data, the selected time range of this data is 1998–2007. Data earlier than 1998 was not selected, mainly because the availability and quality of earlier data were relatively poor and were not particularly important for this study. There are three main reasons for not using the data from 2008 onward: First, the seven-year policy implementation time is sufficient to evaluate the carbon emission effect of the WDS. Second, after many years of policy implementation, many other policies will be present that may interfere with the (relative) policy effect of the WDS. For example, the six central provinces (Shanxi, Anhui, Jiangxi, Henan, Hubei, and Hunan) have also been granted significant preferential policies for the WDS since 2007. Third, the Regulation on the Implementation of the Enterprise Income Tax Law of the People’s Republic of China, which has been in effect since 2008, has greatly changed the enterprise income tax landscape, and will have had a significant impact on the effects of the policy. In addition, the county-level fiscal and tax data in this paper are from the National Fiscal Statistics of Cities and Counties from 1999 to 2008, while the county-level economic and social development indicators are from the China Regional Economic Statistics Yearbook over the same years.

The core explanatory variable of this paper is the interaction between the time dummy variable representing the implementation of the WDS and the county dummy variable representing whether the county under analysis has benefited from the WDS. Based on these selection criteria, the policy treatment group we selected was the counties that benefit from the WDS that are close to the provincial boundary, and the reference group was the counties adjacent to the eastern side of the WDS provincial boundary. Therefore, the main research subject of this paper is 188 counties on the east and west sides of the provincial boundary line of the WDS, including 90 counties on the west side (west assigned value is 1) and 98 counties on the east side (west assigned value is 0. (The data of the counties covered by carbon emission data are missing individually, 3 counties on the west side are missing, 1 county on the east side is missing,.) Other control variables involved in this paper include the logarithm of GDP per capita, the logarithm of the permanent population, urbanization rate, education level, that is, the proportion of middle school students in the population, and the level of financial development expressed by the ratio of the current year’s loan balance to GDP. (According to the needs of this research, the proportion of the secondary industry is a relatively important variable, especially in the robustness test. However, we did not choose this variable mainly due to data quality issues. The data sources we used are consistent with those employed by Lei et al. [19]. One of the problems regarding the data quality of this variable is that a considerable proportion of the industrial output value in the data is higher than the total output value). The Table 1 below shows the descriptive statistics of the main variables, for the period of 1998–2007.

Figure 1 below shows the annual average carbon emission and carbon emission intensity of counties on both sides of the provincial boundary of the WDS. From the perspective of total carbon emissions, the carbon emissions of counties farther to the west are lower. However, after the implementation of the WDS, the carbon emissions of counties bordering the provincial boundaries in the west rose relatively quickly, exceeding the average of counties opposite the provincial boundaries from 2006 onward. With regard to carbon emission intensity, following the implementation of the WDS, the counties east of the border provinces exhibit a downward trend, and the cumulative decline is larger, while the carbon emission intensity of the counties west of the border province with the WDS do not decrease. Significantly, even in 2003 and 2005, there was a relatively large increase in carbon emission intensity, and the overall decline was significantly lower than that of other groups. By 2007, counties west of the WDS boundary which were under the WDS had moved from the group with the lowest carbon intensity to the group with the second highest carbon intensity. These two figures provide a preliminarily demonstration the basic conclusion of this paper, that is, the WDS raises the carbon emission intensity of the counties it covers. Next, this paper will conduct a more in-depth analysis through more rigorous econometric methods.

In addition to the core dependent variables, we have also grouped some other important variables (distinguishing between the east and west before and after the WDS) statistics (see Appendix A). Per capita GDP, financial development level, fiscal income, and other indicators all increase over time. At the beginning of the data collection period, these indicators for western counties were on average slightly lower than neighboring eastern counties. However, the mean difference was not significant. These results are consistent with the findings of Liu et al. [27].

### 4.2. Main Results

As previously mentioned, this paper examines the impact of the WDS on the county carbon emission intensity of China’s western and non-western regions, that is, the net effect of the WDS. A fixed effect model was used to generate estimates. As there are significant differences between the economic development and operation mechanisms of these two types of regions, it is difficult to completely deal with this problem by controlling the relevant variables for each county economy. The fixed effect model is thus a more appropriate and logical choice. Table 2 reports the main regression results. The dependent variable is carbon emission intensity. In addition to the core independent variable of the interaction between policy and western counties, Model 1 adds the dummy variable of policy, and Model 2 adds the dummy variable of year. Some control variables, such as GDP per capita, population, urbanization rate, and education level, were added to Models 3–5. The results of these models show that the carbon emission intensity of the counties in the west has increased significantly after the implementation of the WDS. According to the results of Model 5, after controlling the relevant variables, the WDS policy increased the carbon emissions of the county per 10,000 yuan of GDP by 1.275 tons, which is significant at the level of 1%. According to Model 2, the WDS increased the carbon emissions of western counties per 10,000 yuan of GDP by 0.915 tons.

How can the accuracy of the core independent variable estimates in Models 2 and 5 be judged? After the control variable was added, the estimated coefficient of the core independent variable was significantly improved, while the coefficient value was also improved. Among these control variables, GDP per capita had the greatest impact on the estimated results of core independent variables. Theoretically, per capita GDP is likely to be endogenous to the WDS to some extent. According to the regression results, the WDS has had a certain impact on the per capita GDP of the western counties bordering the provincial boundary. When controlling GDP per capita (Models 3–5), the estimated coefficient of GDP per capita was significantly negative, and the two dummy variables, policy and year, were significantly positive, which had a significant impact on the fitting coefficient of the regression population. This indicates that the growth of GDP per capita and the transformation of the economic development mode are realized concurrently over time, which may be the reason why economic development is accompanied by a “natural” reduction of carbon emission intensity. It can also provide a possible explanation for the larger and more significant estimation coefficient of the core independent variable when the per capita GDP is added as the control variable. In other words, the WDS promotes GDP growth, while the carbon emissions originally caused by the increase in per capita GDP naturally decrease over time.

In Models 2 and 4, we control the dummy variable of year, while in Models 1 and 3, we only control the dummy variable of policy. When only the dummy variable of policy is controlled, the policy variable is significantly negative, which mainly implies that the carbon emission intensity decreases over time based on the year involved in the data. The results of Models 2 and 4 also indicate this is the case. With the time-fixed effect, the coefficients of the dummy variable of years after 2001 are negative and almost significant. This may be attributed to the transformation of China’s overall economic development mode and the improvement of the technology level. (Similar results and consistent interpretations have been reported in many studies, such as those of Lei et al. [19] and Wang et al. [21]).

Next, we will explain the results of other control variables in Model 5. The estimation results show that the greater the size of the population, the lower the carbon emission intensity. This may indicate that counties with higher population sizes tend to have a higher proportion of tertiary or primary industry, with the carbon emission intensity of these two industries being relatively low. The urbanization rate is not significant when per capita GDP is not controlled, but is significantly positive when the per capita GDP and population are both controlled simultaneously. This may indicate that, given the per capita GDP and population, a higher urbanization rate means a higher proportion of secondary industry, which corresponds to a higher carbon emission intensity. As previously cited with regard to per capita GDP, these control variables may to a certain extent be endogenous to the WDS Policy, which may potentially lead to errors.

Table 3 (Columns 1 and 2) reports the regression results when the total carbon emissions are used as the dependent variable. The results show that the interaction between the dummy variable of time when the WDS was implemented, and the dummy variable of the western counties is significantly positive. The total emissions of the western counties increased by 16.4% due to carbon emissions. Columns 3 and 4 report the regression results of per capita carbon emissions as the dependent variable. The results show that the WDS significantly increased per capita carbon emissions.

### 4.3. Robustness Check

#### 4.3.1. Propensity Score Matching Difference-in-Differences Method (PSM-DID)

The propensity score matching (PSM) method can determine whether the sample counties can enter the treatment group by screening the variables, and then conducting a difference-in-differences test after matching the samples according to the propensity scores of these variables. We selected three indicators, GDP per capita, urbanization rate, and local financial proportion, to conduct a logit regression on whether the county would become a western county. We adopted the kernel matching method to match, and then examined the DID results. The test results are shown in Table 4. The treatment effect obtained by the PSM-DID method was 1.166, that is, the WDS has increased the carbon emissions per 10,000 yuan of GDP in western counties by 1.166 tons, which is significant at the 10% level.

To show the PSM matching effect of the treatment group and control group, this paper draws the kernel density function curves before and after propensity score matching for comparative analysis (see Figure 2), and a graph showing the common value range of the propensity score (see Figure 3). As can be seen from Figure 2, the treatment group and control group were unbalanced before the matching. The probability density distribution of the propensity score values of the two groups of samples differed to a certain extent, and there was a possibility of selection bias between the samples of the treatment group and control group. After the nearest neighbor tendency score matching, the nuclear density curves of the treatment group and the control group were very similar, and the probability density distribution of the two groups of samples retained was significantly more balanced than before the matching. The difference between the matched treatment group and control group decreased significantly, indicating that the main characteristics of the two groups of samples after matching were close, and that the sample selection error was controlled. Figure 3 illustrates that the standardized deviation of the variables that affect whether the sample is a western county was greatly reduced after matching. This data clearly demonstrates that the application of the PSM method was more effective.

#### 4.3.2. Parallel Trend Test

The parallel trend test is designed to exclude as much as possible significant differences that the policy role and non-policy role samples have produced prior to the policy implementation. To achieve this goal, we can make a preliminary observation from the time trend chart. As shown in Figure 1, the carbon emission intensity of the border counties in the western neighboring provinces shows an inconsistent trend with that of the eastern groups from 2001 onward. We get the dynamic effect map of the policy by using the conventional event research method and adding the dummy variables before and after the policy (taking the year 2000 as a reference point). The dynamic effect in Figure 4 shows that the coefficient is close to 0 and that while the effect is quite insignificant one and two years before the policy effect, it has been significantly positive since 2002, with the coefficient exhibiting a slight upward trend over time. This may be due to the time lag effect and the cumulative effect of policies on the enhancement of carbon emission intensity. This also provides evidence for the mechanism of the WDS to improve carbon emission intensity: after all, it takes a certain amount of time for industrial enterprises to expand production or build new capacity.

#### 4.3.3. Placebo Test

The purpose of the placebo test is to imagine a sample of “pseudo policy effect” that does not exist, or for which there is sufficient reason to believe does not work. By proving that the treatment effect obtained by these pseudo-treatment groups is significantly different from the actual policy effect, the policy effect caused by random factors can be excluded to a certain extent. We randomly selected 90 counties from the 188 counties included on the east and west sides of the WDS provincial boundary as the “pseudo-treatment group.” Then the same estimation model was used for estimation. This was repeated 500 times, and the core independent variable coefficient and significance of these 500 times was plotted, as shown in Figure 5 and Figure 6. Figure 5 illustrates that most of the core independent variable coefficients over the 500 repeats were near 0, and none of them exceeded 1. The significance is close to 1 as the coefficient is close to 0. Based on these results, it is hard to believe that the benchmark regression results were caused by random factors.

Counties in the east of the provincial boundary of the WDS can be seen as other possible pseudo-treatment groups. We constructed 3 pseudo-treatment groups, including 1 unit in the east (counties adjacent to the eastern side of the reference counties), 2 units in the east (counties adjacent to the eastern side of the reference counties in “unit 1 in the east”), and 3 units in the east (counties adjacent to the eastern side of the reference counties in “unit 2 in the east”). These groups were combined to form a series of specific “pseudo” treatment control groups. The number of these counties in our data was 86, 90, and 81, respectively. Table 5 reports this series of regression results. In Model 1, the county on the east of the provincial boundary was taken as the treatment group, and the county moving one unit eastward was taken as the control group. In Model 2, the counties on the east side of the provincial boundary moved 1 unit eastward as the treatment group, and the counties moved 2 units eastward as the control group. In Model 3, 2 units of counties east of the provincial boundary were taken as the treatment group, and 3 units of counties in the eastern region were taken as the control group. In Model 4, the counties to the east of the provincial boundary were taken as the treatment group, and the counties that moved 2 units eastward were taken as the control group. In Model 5, the counties in the east of China were taken as the treatment group, and the counties that moved 3 units eastward were taken as the control group. In the test results of the five “pseudo treatment” effects, the interaction of policy and time related dummy variables and pseudo treatment dummy variables was not significant. This placebo test can help to explain that the treatment effect obtained in the benchmark model is unlikely to be due to some unknown reason that just caused the difference in carbon emission intensity of neighboring counties around the year 2000.

#### 4.3.4. Change Reference Group

One possible reason to refute the inferences made in this paper is that the above model results may be caused by the special situation of the reference group. To test this idea, we changed the reference group for testing. In addition to the counties close to the east of the provincial boundary involved in the regression model, our dataset also includes the counties that move 1, 2, and 3 units eastward, respectively. We estimated these three groups as reference groups respectively, with the results reported in Table 6. Column 1 shows that the reference group moved one unit county eastward. Compared with the benchmark result (column 2 of Table 2), the interaction coefficient of the western dummy variable and the policy time dummy variable increased from 0.915 to 1.005. That is to say, the carbon emission intensity brought by the WDS to the western counties has also increased slightly, which is also significant at the level of 5%. The reference groups in Columns 2 and 3 respectively moved 2 and 3 units eastward. The core variable coefficient decreased slightly, but was still significant at the 10% level. This shows that our core results are not simply due to the special selection of the reference group, but are robust in this sense. (When the “base reference group” (reference group in 4.2) was replaced by those counties adjacent to the eastern side of counties in “base reference group”, the problem caused by the differences in unobservable factors between the treatment group and the reference group would be more severe since there are many significant differences in east China and west China, so this is not a strict treatment effect estimation, but only a relevent robustness test).

#### 4.3.5. Extension of Observation Time

We gave three reasons why we used data from 1998–2007 rather than more recent data in the previous section. An additional reason is that the observation data in many authoritative studies did not extend into recent years [24,28]. However, it is necessary to report the results over a longer period of time to ensure the robustness of conclusions and better comparability with other studies (e.g., Zhang, et al. [24], Zhang, et al. [40]). Therefore, we supplemented the data to 2015, except for the urbanization rate due to excessive missing values. The results are shown in Table 7. The results are consistent with the basic conclusions of the previous article; the development of the west has significantly improved the carbon emission intensity of western counties, but has not significantly increased the per capita GDP.

#### 4.3.6. External Effects

As mentioned above, after the WDS was implemented, the carbon emission intensity, with particular reference to total carbon emissions, of the counties to the west of the provincial boundary, increased compared with the counties to the east of the provincial boundary. This may be due to the “external effect” of the carbon emission effect of the WDS [40]. That is to say, the increase of carbon emissions in western counties adjacent to the provincial boundary may be due to the transfer of carbon emissions from eastern counties adjacent to the provincial boundary. In other words, if there is an “external effect”, the results obtained in our benchmark model may be directly caused by the migration of carbon emission activities from adjacent areas. If it is due to adjacent migration, it means that the carbon emission intensity of the eastern group, especially the total amount, will be lower and lower as it moves eastward, that is, there would be a significant decline.

After analyzing our data, we did not find this to be the case. First, our previous “pseudo processing reference group” and replacement reference group tests showed that there was no gradual decline of carbon emission intensity of the eastern counties adjacent to the provincial boundary. Second, we took the total carbon emissions as the dependent variable, and created a “pseudo treatment reference group” to replace the reference group. The results are shown in Table 8. Models 1, 2, and 3 show that there was no significant difference in the total carbon emissions of the four groups of counties to the east of the provincial boundary, compared with their neighboring counties after the WDS was implemented. Models 4 and 5 show that the carbon emissions of the eastern counties have increased to a certain extent post WDS compared with the eastern counties that moved 2 units or 3 units eastward. The results of these tests do not support the “externality” hypothesis of increased carbon emissions brought about by the WDS. At the same time, because carbon emissions are mainly the result of the production and investment behavior of enterprises, it is reasonable to speculate that the above results show that there is no significant “neighbor migration” effect of enterprises. This is consistent with the findings of Liu et al. [22], which do not support the main conclusions of Luo et al. [29].

## 5. Discussion

In a large number of studies surrounding the evaluation of the WDS, two important questions have yet to reach a unified conclusion. The first: is the WDS a growth enhancer or a “policy trap”? The other: does the development of the western region negatively affect the nearby regional economy?—that is, is there a negative “external effect” from the policy? Economic policy assessments can have two counterfactual perspectives, one of what would have happened to the treatment group without the policy, and the other of the overall results if the resources to implement the policy were effectively repurposed. Most of the research (such as the literature covered in this article) focuses on the first perspective, while the second perspective is rarely covered, as its difficulty is not comparable. Regarding the first question, Zhang et al. [24] used the PSM-DID method to estimate the provincial panel data and showed that the implementation of the WDS increased the average annual economic growth rate of China’s western region by about 1.5 % after its introduction. Alder et al. [41] used data from a panel of Chinese (prefecture-level) cities from 1988 to 2010 and found that for every 1% decrease in the nominal tax rate, the average TFP of the enterprise production efficiency increased by between 0.38–0.75%, although this effect weakened over time. This result is consistent with the theoretical model of this study, that is, lowering the capital price will result in higher production efficiency until the capital quantity is optimized. The paper also finds that new firms are incentivized to enter the western region, which is consistent with the conclusions of Liu et al. [22].

Although the conclusions of most of the literature are consistent with the basic conclusions of Zhang et al. [24], the research of Liu et al. [27] has obtained diametrically opposed results. Both Zhang et al. [24] and Liu et al. [27] applied DID and PSM-DID methods to estimate the economic growth effect of the WDS on prefecture-level cities. Their results stated that the WDS did not promote the growth of the western region, and even created an “economic trap” instead. The contrast between the two documents is also reflected in other aspects such as, for example, the regression results. We believe that there is no universally precise relationship between government spending and economic growth. However, these two studies conclude that government spending has significantly opposite effects on regional economic growth in a similar time zone, which may be more about government spending projects at different levels and the factors determining them. The scale of government spending (as a percentage of GDP) in the western region is low. To a certain extent, it is endogenous to the WDS (development and large-scale aid projects and infrastructure construction are not included in the government expenditure at the same level and are included in the GDP). It should be noted that this empirical conclusion might be considered debatable because the dependent variable in this paper is a prefecture-level city rather than county-level GDP, or per capita GDP. In addition, when using the DID method to test, this paper uses a random effect model, and there is no initial value of the control variable, which may cause serious biases caused by unobservable factors in the estimation results.

The present paper attempts to bridge the above two groups of opposing results. We also tested GDP per capita as dependent variables, as shown in Table 9.

Using the basic DID test, we found that the WDS seems to have increased the per capita GDP of the western counties (as shown in Columns 1 and 2), but there are problems with the robustness of this conclusion. When changing the reference group (Columns 3, 4, and 5) or using the aforementioned PSM-DID method, the results show that the WDS has not significantly increased the per capita GDP of the western counties. In all tests, we did not record a significant negative growth effect. How then can a more robust increase in carbon emissions and a less significant, less robust growth effect both be explained? The simple model that we built mainly emphasizes that the WDS will increase the capital input and output of enterprises that receive preferential treatment. However, if we take a broader view, taking into account factors such as the interference of these preferential policies in the human capital market, the relative price distortion between different industries, and the fact that non-market behaviors caused by the WDS other than tax incentives may reduce the growth effect, we can explain our empirical results. To a certain extent, it also bridges the gap between the two opposing schools of thought represented in the existing literature. The research of Alder et al. [41] shows that regional economic policies have increased human capital expenditure, however, this study finds that the WDS has not increased the population at all (the coefficient is not significantly negative) in the border counties of neighboring provinces in the west. Higher human capital spending did not attract a larger population, possibly because the labor market became less efficient.

When evaluating the effects of policies through different levels of economic performance, another issue worth mentioning is the extrapolation validity of the conclusions. Most of the previous research in this area has attempted to evaluate the overall effect of the WDS [18,42,43,44,45,46]. However, the regional economy is the sum of the county-level, prefecture-level, and provincial economies, and each level will give priority to the effect of the policy at its level in the evaluation. In China, a large country with obvious regional differences, it is crucial to determine the key driving force of regional carbon emission intensity, in order to make reasonable adjustments to the regional economic development strategy. Besides, the regional economy has obvious agglomeration characteristics. The economic performance of a city or county cannot completely predict the economic performance of the entire region, and the WDS may create heterogeneous outcomes at different levels of the economy. Lu and Deng [23] provide evidence that while the WDS has played a significant role in promoting the transformation and upgrading of the industrial structure of high administrative level cities in the west, it has not proven significant for cities in the western region in general.

On the second question, regarding whether the development of the western region negatively affected the nearby regional economy, there are two main reasons why external effects will affect the robustness of policy effects. First, this factor is itself an important aspect of economic policy evaluation. Second, when comparing with neighboring regions to evaluate the effects of the WDS, the external effects will inevitably have an impact on the robustness. Liu et al. [27] applied the triple difference method to study the changes in the number of classified enterprises at the county level near the western provincial boundary before and after the implementation of the WDS. The results showed that the WDS significantly increased the number of enterprises in the western counties, while the number of enterprises in the counties to the east of the provincial boundary was not significantly affected. This contradicts the hypothesis of Luo et al. [29], who studied the relationship between local policies and economic development. Research results of Luo et al. [29] showed that enterprises in economically developed areas would migrate to less developed areas. From the estimation results above (Section 4.3.6. External Effects), the total amount and intensity of carbon emissions does not lead to migration effects, and carbon emissions can directly reflect the business activity of enterprises. Therefore, reasonable speculation based on these results is more consistent with the conclusion of Liu et al. [27]. That is, the WDS has not produced negative external effects on neighboring areas, and the possibility of enterprises moving nearby is not significant. This is not difficult to understand. County-level enterprises are usually small in scale, and because their production and operation depend on local social relations, they often take root locally. Although the transportation cost of moving to a neighboring province may be relatively low, in general, it is not necessarily advantageous to move to the west of China to set up new enterprises compared with enterprises with larger scales and farther away.

## 6. Conclusions

This study constructs a duopoly output decision-making model to illustrate that tax reduction, the main policy of the WDS, will improve the optimal capital stock and output of enterprises, thus improving enterprises’ carbon emission level and intensity. The DID method is then used to compare the impact of the WDS on carbon emissions at the county level on both sides of the WDS policy boundary, and conduct various robustness tests. Our estimated results show that the WDS has significantly increased the carbon emission intensity and total carbon emissions of western counties, with these results having passed multiple robustness tests. These findings indicate that the WDS has increased the high carbon emission behavior of China’s western region, which may be partly due to the attraction of enterprises with high carbon emission intensity.

Based on an extensive review of the existing theory and relevant literature on this subject, combined with empirical data, this study has found that the WDS has no significant and stable positive effect on economic growth at the county level. However, no evidence of a policy trap was found either. The empirical evidence of this paper also implies that the WDS has not produced obvious “negative externalities” caused by nearby migration. The core policy implication of this study may be that the practice of regional economic policies is likely to lead to an increase in carbon emissions. Given that emission reduction is now a major national strategy, the impact of carbon emissions and regional economic policies should be considered on a national rather than regional level. Now, when formulating regional economic policies, such as preferential arrangements and financial support, carbon emission targets and environmental costs should be considered, and relevant targets should be included. In some cases, the number of carbon emissions can also become a way to support regional development. When a national carbon trading market is formed, this regional economic policy can be achieved in a more market-oriented manner.

The inadequacies of this paper mainly lie in the fact that economic policymaking is a systematic project. The evaluation of policy is also a systematic project, which requires multi-level and multi-faceted theoretical and empirical research. Therefore, it is necessary to further explore different levels (provinces, cities, counties, enterprises), and use different methods, approaching from different perspective to obtain more detailed and reliable conclusions. The local point of view must be considered in order to achieve more comprehensive and stable conclusions in general, which is also a key directive for future research. It is undoubtedly a major challenge for the Chinese government to reduce CO_2_ emissions and achieve their climate change mitigation targets with a 40% reduction of carbon intensity by 2025, without compromising any of their development goals. It is crucial to consider, and successfully integrate both regional economic development and climate change mitigation strategies. Future research should pay more attention to balancing the effects of economic development and environmental protection policies simultaneously.

## Figures and Tables

**Figure 1 ijerph-20-02669-f001:**
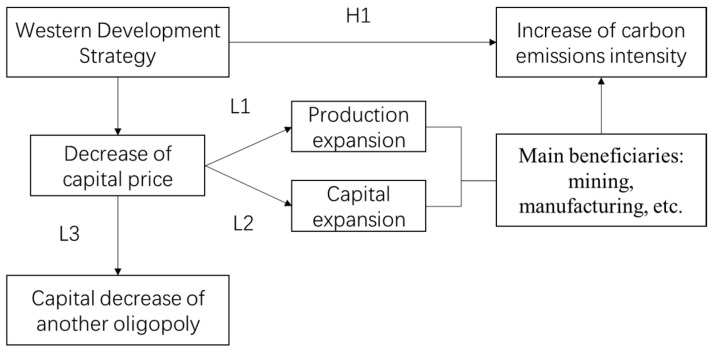
Research framework.

**Figure 2 ijerph-20-02669-f002:**
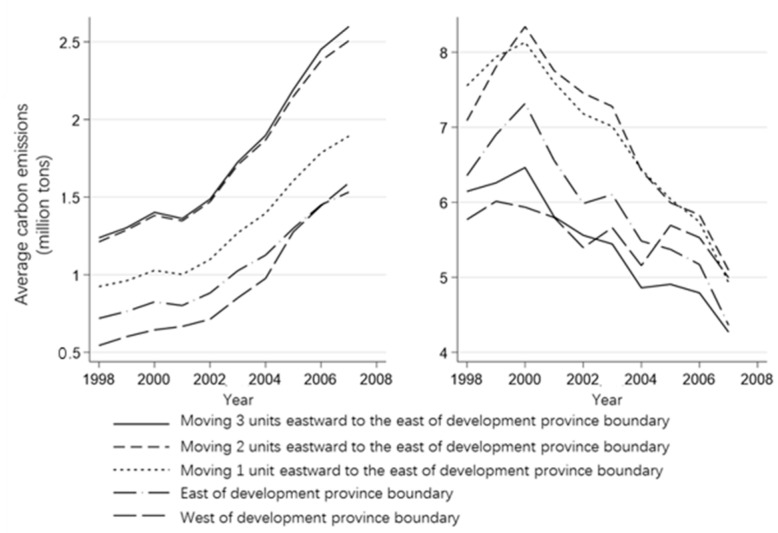
Average carbon emission intensity and average carbon emission of five types of counties on both sides of the development province boundary in each year.

**Figure 3 ijerph-20-02669-f003:**
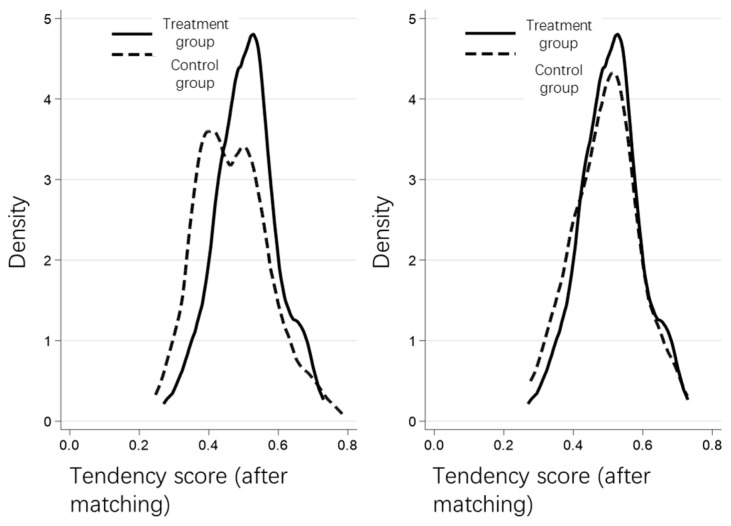
Kernel density function curve.

**Figure 4 ijerph-20-02669-f004:**
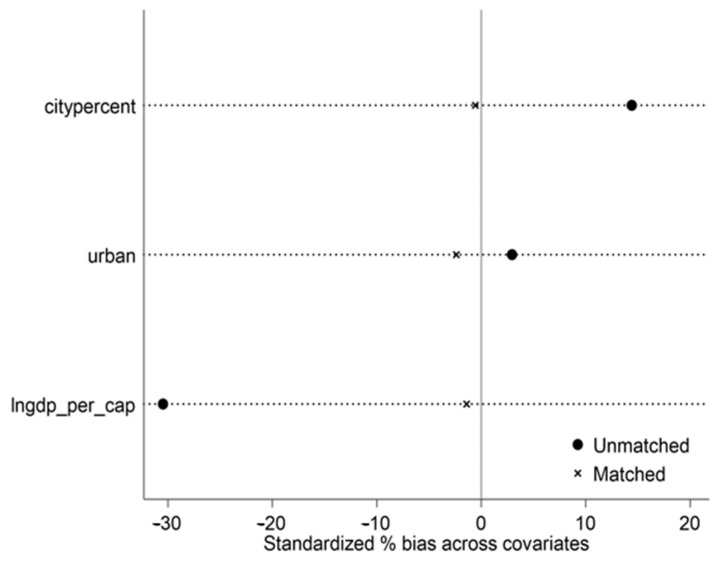
Common value range diagram.

**Figure 5 ijerph-20-02669-f005:**
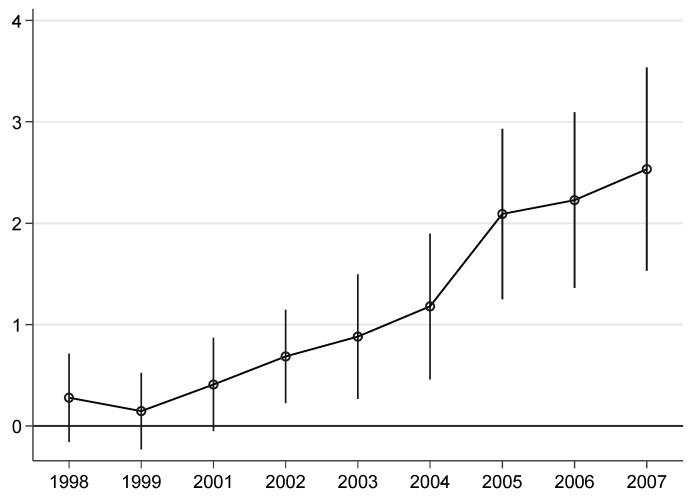
Parallel trend test.

**Figure 6 ijerph-20-02669-f006:**
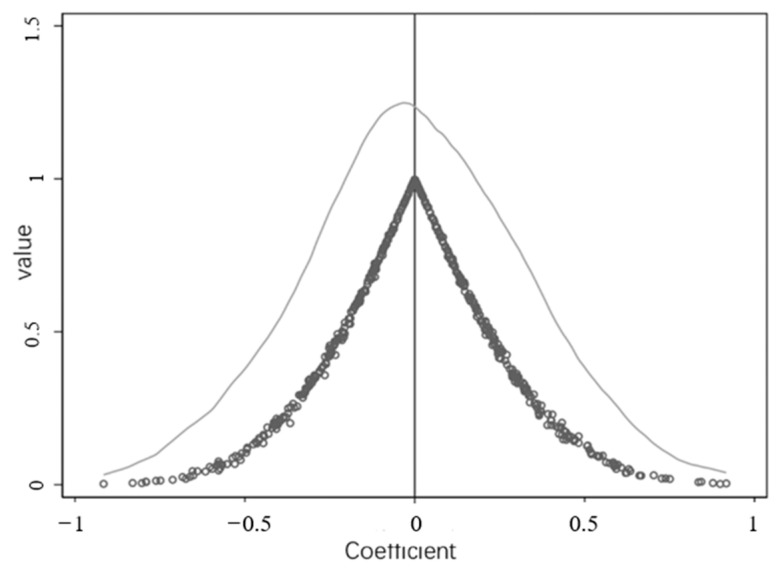
Coefficients and *p*-values of placebo tests for randomized treatment groups. Note: The circles represent the scatterplot of *p*-values and estimated coefficients; the lines represent the kernel density estimation of the estimated coefficients.

**Table 1 ijerph-20-02669-t001:** Descriptive statistics of main variables.

Variable	Number of Samples	Mean	Standard Deviation	Minimum	Maximum Value
West	4.45	−0.94	1.40	−3	1
Year	4.45	2003	2.87	1998	2007
Emm (carbon emissions)	4.36	1.34	1.27	0.001	11.78
Emmden (carbon intensity)	4.33	6.16	5.98	0.06	55.76
Lngdp (ln ten thousand yuan)	4.41	12.20	0.98	9.00	15.20
Lnpop (ln population)	4.40	3.60	0.67	1.61	5.12
Urban (urbanization rate)	4.34	0.21	0.15	0.01	1
Edu (educational level)	4.40	0.06	0.02	0.01	0.45

Note: Carbon emission unit: million tons; carbon emission intensity unit: ton/10,000 yuan GDP.

**Table 2 ijerph-20-02669-t002:** Statistical analysis process.

	(1)	(2)	(3)	(4)	(5)
Variable	Carbon Intensity				
Interactive term	0.918 **	0.915 **	1.137 ***	1.404 ***	1.275 ***
	(0.434)	(0.434)	(0.369)	(0.333)	(0.309)
GDP per capita			−3.105 ***	−6.784 ***	−7.012 ***
			(0.406)	(0.719)	(0.699)
Lnpop					−12.462 ***
					(2.504)
Urban					3.632 **
					(1.701)
Edu					−11.254 ***
					(4.078)
Individual fixed effects	√	√	√	√	√
year fixed effect		√		√	√
post	−1.331 ***		0.203		
	(0.336)		(0.187)		
Constant	6.405 ***	6.081 ***	31.323 ***	60.333 ***	105.098 ***
	(0.154)	(0.165)	(3.368)	(5.808)	(12.719)
Observations	1826	1826	1819	1819	1788
R-squared	0.053	0.093	0.295	0.474	0.515
Number of counties	184	184	184	184	184

Note: (1) The robust standard errors of clusters (to counties) are in parentheses; (2) *** indicates the significance level of 1%; ** indicates the significance level of 5%.

**Table 3 ijerph-20-02669-t003:** WDS and Total Carbon Emissions and Per Capita Carbon Emissions.

	(1)	(2)	(3)	(4)
Variable	Ln Carbon Emissions		Carbon Emissions Per Capita	
Interactive term	0.164 ***	0.164 ***	0.007 **	0.007 **
	(0.038)	(0.037)	(0.003)	(0.003)
Individual fixed effects	√	√	√	√
Year fixed effect	√		√	
Large development Dummy variables		√		√
Constant	−1.021 ***	−0.932 ***	0.020 ***	0.021 ***
	(0.016)	(0.013)	(0.001)	(0.001)
Observations	1840	1840	1822	1822
R-squared	0.835	0.417	0.361	0.140
Number of counties	184	184	184	184

Note: (1) The robust standard errors of clusters (to counties) are in parentheses; (2) *** indicates the significance level of 1%; ** indicates the significance level of 5%.

**Table 4 ijerph-20-02669-t004:** PSM-DID test results.

	Control Group (Pre-WDS)	Treatment Group (Pre-WDS)	Treatment Group (Pre- WDS)— Control Group (Pre-WDS)	Control Group (Post- WDS	Treatment Group (Post- WDS)	Treatment Group (Post- WDS)— Control Group (Post- WDS	Difference-in-Differences Test Results
Carbon emission Density	7.873	5.906	−1.964	6.262	5.461	−0.801	1.166
Standard error			0.521			0.342	0.623
T value			−3.78			2.34	1.87
Salience			0.000			0.019	0.061

Note: The number of samples entered into difference-in-differences is 1799, and the R-squared is 0.02.

**Table 5 ijerph-20-02669-t005:** “Pseudo treatment-control group” test.

Variable	(1)	(2)	(3)	(4)	(5)
	Eastside County vs. Eastside County Moved 1 Unit Eastward	Eastside Counties Moved 1 Unit Eastward vs. Eastside Counties Moved 2 Units Eastward	Eastside Counties Moved 2 Units Eastward vs. Eastside Counties Moved 3 Units Eastward	Eastside Counties vs. Eastside Counties Moved 2 Units Eastward	Eastside Counties vs. Eastside Counties Moved 3 Units Eastward
	Carbon Intensity				
“Pseudo-handling” interaction	0.089	−0.159	−0.102	−0.069	−0.171
	(0.503)	(0.499)	(0.438)	(0.472)	(0.443)
Individual fixed effects	√	√	√	√	√
Year fixed effect	√	√	√	√	√
Constant	6.925 ***	7.331 ***	6.655 ***	6.771 ***	6.263 ***
	(0.180)	(0.168)	(0.138)	(0.144)	(0.156)
Observations	1768	1712	1674	1824	1730
R-squared	0.142	0.149	0.144	0.143	0.142
Number of counties	178	172	169	184	175

Note: (1) The robust standard errors of clusters (to counties) are in parentheses; (2) *** indicates the significance level of 1%.

**Table 6 ijerph-20-02669-t006:** Tests for changing the reference group.

	(1)	(2)	(3)
	Westside Counties vs. Westside Counties Moved 1 Unit Eastward	Westside Counties vs. Westside Counties Moved 2 Units Eastward	Westside Counties vs. Westside Counties Moved 3 Units Eastward
Variable	Carbon Intensity		
Interactive term (Control group moved east)	1.005 **	0.846 *	0.744 *
	(0.464)	(0.429)	(0.397)
Individual fixed effects	√	√	√
Year fixed effect	√	√	√
Constant	6.591 ***	6.447 ***	5.900 ***
	(0.187)	(0.152)	(0.162)
Observations	1714	1770	1676
R-squared	0.093	0.092	0.087
Number of counties	172	178	169

Note: (1) The robust standard errors of clusters (to counties) are in parentheses; (2) *** indicates the significance level of 1%; ** indicates the significance level of 5%; * indicates the significance level of 10% level..

**Table 7 ijerph-20-02669-t007:** Tests for changing the observation time.

	(1)	(2)	(3)	(4)	(5)
Variable	Carbon Intensity			Lngdp_per_cap	
Interactive term	1.363 **	1.588 ***	2.125 ***	0.087	0.093 *
	(0.569)	(0.506)	(0.439)	(0.057)	(0.052)
GDP per capita		−2.734 ***	−5.924 ***		
		(0.552)	(0.846)		
Lnpop			−5.808 ***		−0.850 ***
			(0.902)		(0.045)
Edu			−7.333 **		−0.507
			(3.649)		(0.606)
Individual fixed effects	√	√	√	√	√
Year fixed effect	√	√	√	√	√
Constant	6.018 ***	27.928 ***	73.829 ***	8.016***	10.987 ***
	(0.235)	(4.532)	(9.775)	(0.024)	(0.159)
Observations	2991	2990	2908	3027	2944
R-squared	0.256	0.359	0.515	0.969	0.984
Number of xzdm	184	184	184	188	188

Note: (1) The robust standard errors of clusters (to counties) are in parentheses; (2) *** indicates the significance level of 1%; ** indicates the significance level of 5%; * indicates the significance level of 10% level.

**Table 8 ijerph-20-02669-t008:** Comparison of the impact of the WDS on the total carbon emissions of various counties on the east side.

	(1)	(2)	(3)	(4)	(5)
	Eastside County vs. Eastside County Moved 1 Unit Eastward	Eastside County Moved 1 Unit Eastward vs. Eastside County Moved 2 Units Eastward	Eastside Counties Moved 2 Units Eastward vs. Eastside Counties Moved 3 Units Eastward	Eastside County vs. Eastside County Moved 2 Units Eastward	Eastside County vs. Eastside County Moved 3 Units Eastward
Variable	Ln Carbon Emissions				
“Pseudo-treatment” interaction effects	0.017	0.016	0.013	0.033 *	0.046 ***
	(0.020)	(0.018)	(0.014)	(0.017)	(0.016)
Individual fixed effects	√	√	√	√	√
Year fixed effect	√	√	√	√	√
Constant	−0.615 ***	−0.242 ***	−0.054 ***	−0.449 ***	−0.439 ***
	(0.009)	(0.009)	(0.008)	(0.009)	(0.008)
Observations	1780	1720	1690	1840	1750
R-squared	0.921	0.929	0.953	0.932	0.940
Number of counties	178	172	169	184	175

Note: (1) The robust standard errors of clusters (to counties) are in parentheses; (2) *** indicates the significance level of 1%; * indicates the significance level of 10% level.

**Table 9 ijerph-20-02669-t009:** **Western Development and GDP per capita (tested by different groups)**.

	(1)	(2)	(3)	(4)	(5)
	West County vs. East County	West County vs. East County	Counties on the West Side vs. 1 County on the East Side	Counties on the West Side vs. 2 Counties on the East Side	Counties on the West Side vs. 1 County on the East Side
Variable	GDP Per Capita				
interactive term	0.082 *	0.084 **	0.060	0.062	0.035
	(0.042)	(0.042)	(0.039)	(0.038)	(0.038)
individual fixed effects	√	√	√	√	√
year fixed effect		√	√	√	√
large development dummy variables	√				
Constant	7.997 ***	7.912 ***	8.049 ***	8.131 ***	8.182 ***
	(0.013)	(0.019)	(0.017)	(0.017)	(0.017)
Observations	2.039	2.039	1.745	1.779	1.687
R-squared	0.419	0.778	0.789	0.803	0.805
Number of counties	188	188	176	180	171

Note: (1) The robust standard errors of clusters (to counties) are in parentheses; (2) *** indicates the significance level of 1%; ** indicates the significance level of 5%; * indicates the significance level of 10% level.

## Data Availability

Data of carbon emission used in this paper can be found at https://www.nature.com/articles/s41597-020-00736-3 (accessed on 29 January 2022). Data of counties can be found in China Statistical Yearbook for Regional Economy (1999–2016).

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
