# Peer review of "The Impact of Location-Based Tax Incentives and Carbon Emission Intensity: Evidence from China’s Western Development Strategy"

_ijerph, 2023, doi:10.3390/ijerph20032669_

Round 1

Reviewer 1 Report

In my humble opinion:
The topic is timely, namely the primary research objective of finding out how China's Western Development Strategy (WDS) affects the behavior of economic actors in the region and how this in turn affects overall carbon emissions and related economic performance. The authors correctly identified the knowledge gap. The WDS aims to reverse the unbalanced economic development of China's eastern and western regions through a series of preferential policy support measures, the most important of which is a substantial income tax exemption for state-supported enterprises. To show the above effects in detail, the authors developed a duopoly output decision model. After presenting the mathematical proof, the authors define three lemmas in service of the main hypothesis: "The WDS will increase carbon emission intensity and total carbon emissions in the western region." The framework of the problem is shown graphically in Figure 1. This manuscript is clear and well structured. The methodological work on which the original conclusions are based is good. The empirical evidence in this paper also suggests that the WDS has not produced obvious "negative externalities" from close migration. An important policy implication of this study may be that regional economic policy practices are likely to increase carbon emissions.

I had the greatest concerns about the data used by the authors. Although the authors detail why they use data from 1998-2007 and also give three reasons why they do not use more recent data, I would leave that decision up to the editor if you feel that the data used is outdated for this type of journal.

Author Response

Response:

Thank you for the opportunity to revise our manuscript. We have revised the manuscript based on each of your comments. Despite the limited time for revision, we have made every effort to substantively address the issues raised by the reviewers’ comments, and we believe the resulting manuscript has been significantly improved as a result.

Thank you very much for your positive comments and doubts about the observation time of data used (span). We have given three reasons why we do not use more recent data but data from 1998-2007 in the previous section.”First, the seven-year policy implementation time is sufficient to evaluate the carbon emission effect of the WDS. Second, after many years of policy implementation, many other policies will be present that may interfere with the (relative) policy effect of the WDS. For example, the six central provinces (Shanxi, Anhui, Jiangxi, Henan, Hubei, and Hunan) have also been granted significant preferential policies for the WDS since 2007. Third, the Regulation on the Implementation of the Enterprise Income Tax Law of the People's Republic of China, which has been in effect since 2008, has greatly changed the enterprise income tax landscape, and will have had a significant impact on the effects of the policy. In addition, the county-level fiscal and tax data in this paper are from the National Fiscal Statistics of Cities and Counties from 1999 to 2008, while the county-level economic and social development indicators are from the China Regional Economic Statistics Yearbook over the same years.. And we have added the forth reason for further explain. For details, please refer to p.17 of the revised paper. It is reproduced below for your convenience.

“4.3.5 Extension of observation time

We give three reasons why we used data from 1998-2007 rather than more recent data in the previous section. An additional reason is that the observation data in many authoritative studies did not extend into recent years [24, 27]. However, it is necessary to report the results over a longer period of time to ensure the robustness of conclusions and better comparability with other studies (e.g. Zhang, et al. [24], Zhang, et al. [40]). Therefore, we supplemented the data to 2015, except for the urbanization rate due to excessive missing values. The results are shown in Table 7. The results are consistent with the basic conclusions of the previous article; the development of the west has significantly improved the carbon emission intensity of western counties, but has not significantly increased the per capita GDP.

Table 7. Tests for changing the observation time

(1)

(2)

(3)

(4)

(5)

Variable

Carbon intensity

Lngdp_per_cap

Interactive term

1.363**

1.588***

2.125***

0.087

0.093*

(0.569)

(0.506)

(0.439)

(0.057)

(0.052)

GDP per capita

-2.734***

-5.924***

(0.552)

(0.846)

Lnpop

-5.808***

-0.850***

(0.902)

(0.045)

Edu

-7.333**

-0.507

(3.649)

(0.606)

Individual fixed effects

Year fixed effect

Constant

6.018***

27.928***

73.829***

8.016***

10.987***

(0.235)

(4.532)

(9.775)

(0.024)

(0.159)

Observations

2,991

2,990

2,908

3,027

2,944

R-squared

0.256

0.359

0.515

0.969

0.984

Number of xzdm

184

184

184

188

188

REFERENCE

[24] Zhang, C.; Zhou, B.; Wang, Q. Effect of China's western development strategy on carbon intensity. J. Clean. Prod. 2019, 215, 1170-1179.

[27] Shao, S; Qi, Z. Energy exploitation and economic growth in Western China: An emprical analysis based on the resource curse hypothesis. Front. Econ. China 2009, 4, 125-152.

[40] Zhang, C.; Zhao, Z.; Wang, Q. Effect of Western Development Strategy on carbon productivity and its influencing mechanisms. Environ. Dev. Sustain. 2022, 24, 4963-5002.

Reviewer 2 Report

Title: The impact of location-based tax incentives and carbon emission intensity: evidence from China's Western Development Strategy

This study aims to determine whether there has been a change in the carbon intensity of emissions in the regions included in China's Western Development Strategy (WDS). As far as we know, the normative causal inference technique has not been used to thoroughly investigate how this economic growth goal would affect carbon emissions. Their research adds to the growing body of literature analyzing the effects of location-based regulations worldwide and has implications for carbon emission-related policies. Using information collected from 188 cities on the eastern and western sides of China's WDS between 1998 and 2007, they applied the difference-in-difference method to explain the correlation between tax cuts connected to the WDS and carbon emissions. Our research indicates that the WDS has caused a significant increase in the intensity of carbon emissions in the western counties of China. Although the WDS has not had a notable positive influence on economic growth in the participating counties, the authors found no indication of a policy trap effect. There is no evidence that economic activity connected to the WDS has resulted in negative carbon emission externalities in the eastern counties located east of the provincial line.

1#Structure and Style of the proposal:  This paper is organized and clear, and the content of the proposal is appropriate.

there are multiple typos, grammar problems, and issues

Every paragraph contributes to the argumentation of the proposal

2# Methodology and conclusion

-The author presented very well the methodology, and the results were explained. Also, the Author briefly highlights the key study findings along with managerial implications.

Author Response

1#Structure and Style of the proposal:  This paper is organized and clear, and the content of the proposal is appropriate.

there are multiple typos, grammar problems, and issues

Every paragraph contributes to the argumentation of the proposal

Response:

Thank you very much for pointing out this, and we apologize for typos and grammar problems. In response to your comments, we have completely revised the manuscript and all revised parts are marked in red. We have also asked for a proofreading service for the manuscript and we believe the resulting manuscript has been significantly improved as a result.

2# Methodology and conclusion

-The author presented very well the methodology, and the results were explained. Also, the Author briefly highlights the key study findings along with managerial implications.

Response:

Thank you very much for your comments and agree our contributions like this.

Reviewer 3 Report

The research aims to explore the impact of location-based tax incentives on carbon emissions, but the implementation of the Western Development Policy is used as a policy shock in the empirical analysis. The article should give more explanation to prove that the empirical results is due to tax incentives rather than other incentives. Also, the paper should show the changes in taxation in the western region before and after the Western Development Policy, and the differences in tax rates between the western region and the east-central region before and after the policy.

Author Response

Response:

We would like to thank you for your invaluable comments. Indeed, we should demonstrate the role of tax incentives in the western development strategy, and show the effects of policies through more specific data. First of all, WDS has significantly reduced the income tax of eligible enterprises in the western region (from 25% to 15%, with a decrease of 54%), which is a very significant preference. The research of Luo et al. [29] studies tax changes in China and the result shows that WDS has reduced corporate income tax in the western region by 39.5%. Besides, according to the prior research, a large number of studies on WDS have emphasized that the preferential tax policy is a key component of the series of policies for the Western Development. Liu et al. [28] also interpreted the western development policy as tax preference. In addition, we use tax preference as the title, which is actually to emphasize the impact of tax preference as the main policy of WDS on carbon emission intensity, not to emphasize that only tax policy will affect carbon emission. In our research model, tax preference is regarded as a factor that can significantly reduce the capital price. If other policies can reduce the price of capital, it also conforms to the theoretical logic of this paper. To make this more clear, we have revised the wording of the manuscript, especially the research framework (Figure 1) on page 7. It is reproduced below for your convenience.

If the reviewer deems it necessary, we can change the title to "Regional Development Strategy and carbon emission intensity: evidence from China's Western Development Strategy". We believe that this will not significantly change the main purpose and contribution of this study.

Figure 1. Research framework.

REFERENCE

[28] Liu, Z.; Wu, H.; Wu, J. Location-based tax incentives and entrepreneurial activities: evidence from Western Regional Development Strategy in China. Small Bus. Econ. 2019b, 52, 729-742.

[29] Luo, M; Fan, Z.; Chen, C. Redistribution Effect of Regional Tax Preferential Policies——Evidence from the Western Development Strategy. Chin. Ind. Econ. 2019, 2, 61-79.

Round 2

Reviewer 3 Report

The author has made changes based on the comments.